# The Effect of Correlated Color Temperature and Illumination Level of LED Lighting on Visual Comfort during Sustained Attention Activities

Xiaoyun Fu [1], Di Feng [2,*], Xu Jiang [1] and Tingting Wu [1]

1 Department of Industrial Design, School of Design and Architecture, Zhejiang University of Technology, Hangzhou 310023, China
2 Institute of Industrial Design, Zhejiang University of Technology, Hangzhou 310023, China
* Correspondence: fengdi@zjut.edu.cn; Tel.: +86-186-5886-6340

**Abstract:** LED lighting has been widely used in various scenes, but there are few studies on the impact of LED lighting on visual comfort in sustained attention tasks. This paper aims to explore the influence of correlated color temperature (CCT) and illuminance level in LED lighting parameters on human visual comfort. We selected 46 healthy college students (23 male and 23 female). The ages ranged from 22 to 26 years old (average age was 24.2 years). Electroencephalogram (EEG) signals, sustained attention to response test (SART) parameters and subjective evaluation parameters of subjects performing sustained attention tasks under LED lighting were obtained. The results under different conditions were compared, and the effects of experimental lighting parameters on visual comfort were discussed. The results showed that the LED lighting with CCT of 3300 K and illuminance of 300 lx was more comfortable than other combined conditions. In the subjective perception of subjects, 4000 K CCT also had good visual comfort evaluation and caused good task performance. Therefore, our study showed that in sustained attention tasks, when LED lighting conditions were CCT of 3300 and 4300 K and illuminance level was 300 lx, the visual comfort of the subjects was better.

**Keywords:** electroencephalogram (EEG); sustained attention; LED lighting; visual comfort

## 1. Introduction

Lighting is important for human activities. A good lighting environment provides us with an environment in which visual tasks can be performed efficiently and accurately without causing visual fatigue and discomfort [1]. Compared to traditional lighting, LED lighting is able to achieve a wider range of correlated color temperature (CCT), easily providing higher performance with better visual comfort and preference [2]. In addition, LED has become the mainstream lighting for high luminous efficiency, energy saving and easier control of color, correlated color temperature, illuminance level and other parameters.

There are many factors related to LED lighting that affect human cognitive activities, such as color rendition, glare, illuminance level, luminance uniformity, CCT, etc. [3]. CCT and illuminance level are among some of the most important characteristics to human cognitive activities [4]. Moreover, their values are adjustable by users, which are very important parameters in the actual use of LED lighting tools. Kruithof [5] studied the perception of the combination of CCT and illuminance level and found that people preferred the combination of high CCT with high illuminance level and low CCT with low illuminance level. However, his research was limited by the characteristics of the lighting sources at that time, which constrained the applications of LED lighting. Manav et al. [6] studied the influence of illuminance level and CCT on workers, finding that 2000 lx illuminance level had a better feeling of comfort than 500 lx. It also found that the CCT at 4000 K produced a better perception of comfort than 2700 K, which produced more

comfort-related evaluations than 4000 K. In Dangol et al. [7,8], observers preferred 4000 to 6500 K at illuminance levels of 500 lx. In addition, the study of Wang et al. [9] showed that CCT had a significant impact on individuals' subjective comfort and personal preference. The comfort preference of CCT and illuminance level varied in different activities performed by the subjects [10,11]. Liang et al. [12,13] simulated the effects of CCTs and illuminance levels in different light conditions on visual performance and attention level of subjects as well as occupant comfort when passing through thermochromic windows. The results showed that visual acuity was higher under higher CCT, yet a lower CCT was still preferred by subjects and more natural, and acceptable lighting conditions were found in bronze-tinted conditions. The above studies indicated that CCTs and illuminance levels can affect people's visual task performance and comfort. However, due to the small adjustable range of CCT and illuminance level in actual reading, writing and other sustained attention activities, the guidance of lighting tools lacks systematic consideration. The considerations for perceived activities are not sufficient, especially in sustained attention. Therefore, it is crucial to study the influence of CCT and illuminance level of LED lighting on visual comfort in sustained attention activities, so as to improve our efficiency and satisfaction in learning, working and other activities.

Using subjective scales to assess visual comfort while using objective measures to reach more accurate conclusions was a recommended method [14]. Recent advances in psychometric technology, particularly wearable sensor technology, have made it possible to achieve objective measurements of subjects' emotional and physiological states during their exposure to the environment, thereby helping to improve the design process to create environments that meet human needs [15]. In particular, through electroencephalogram (EEG) technology, it is possible to non-invasively measure brain activity, to obtain neurophysiological data independent of individual control, and thus to study neuro-related factors of various cognitive processes, such as visual comfort [16,17]. At present, EEG is widely used for subjective perception and cognitive tasks and can be used as an objective indicator to support traditional subjective perception and task assessment methods [18]. According to the International Federation of Societies for Electroencephalography and Clinical Neurophysiology, EEG bands are divided by frequency, from low to high: δ waves (0–3.5 Hz), θ waves (4–7 Hz), α waves (8–12 Hz), β waves (13–30 Hz), and γ waves (>31 Hz). Among them, α waves occur when people are awake, quiet, calm, stable, and focused. Frey et al. [19] showed that for visual comfort in uncomfortable conditions, signal power decreased significantly in the α band. In addition, Giulia et al. [20] found in their study that visual comfort correlated with α wave signal power, and the visual perception with lower comfort often corresponded to lower α wave signal power. Therefore, the signal power of α waves can be used as a characteristic representation of visual comfort.

The purpose of this study is to explore the relationship of EEG signals, SART, and subjective evaluation, to test the effects of different CCTs and illuminance levels on visual comfort when users use LED lighting for cognitive activities that require us to have longer attention, and to determine the best combination of CCT and illuminance levels. For instance, when reading or writing with an LED table lamp with adjustable CCT and illuminance levels, we can adjust it to the right parameters to help maintain our attention for longer periods without causing visual discomfort. It is expected to help users to achieve the best visual comfort while completing sustained attention tasks, so as to improve their satisfaction with work and study. In addition, the influence of the combination of CCT and illuminance level on the vision of users, especially teenagers, is an important issue for LED lighting manufacturers in the current market. Therefore, it is of great significance for designers of LED lighting equipment to explore the influence of CCT and illuminance level on the visual comfort of sustained attention activities.

## 2. Materials and Methods

This is an experimental study, not a purely theoretical one. We explored the visual comfort caused by LED lighting for daily activities. The research method combined physiological signal measurement with a subjective scale, which has been proven to be rigorous. The theoretical framework of this paper is as follows.

In the study of related theories, this paper verified the theoretical feasibility of the thesis by citing two aspects. First, the paper cited the correlation between CCT, illuminance level and virtual comfort level to prove that there is a significant correlation between them. Second, the paper cited the correlation between visual comfort and sustained attention to prove that comfort level and sustained attention are strongly correlated. Next, we hypothesized that changes in CCT and illuminance levels in LED lighting would not cause changes in visual comfort during sustained attention activities. Then, we selected the daily value range of CCT and illuminance level, graded them and combined them as experimental variables. We then collected the EEG signals of the subjects in the sustained attention to response test (SART) tasks and extracted the features of the frequency band related to visual comfort. At the same time, after each group of experiments, subjects would conduct subjective visual comfort evaluation. Finally, we synthesized the results of objective physiological signals and the results of subjective scales as the basis for the conclusion of this paper. (as shown in Figure 1)

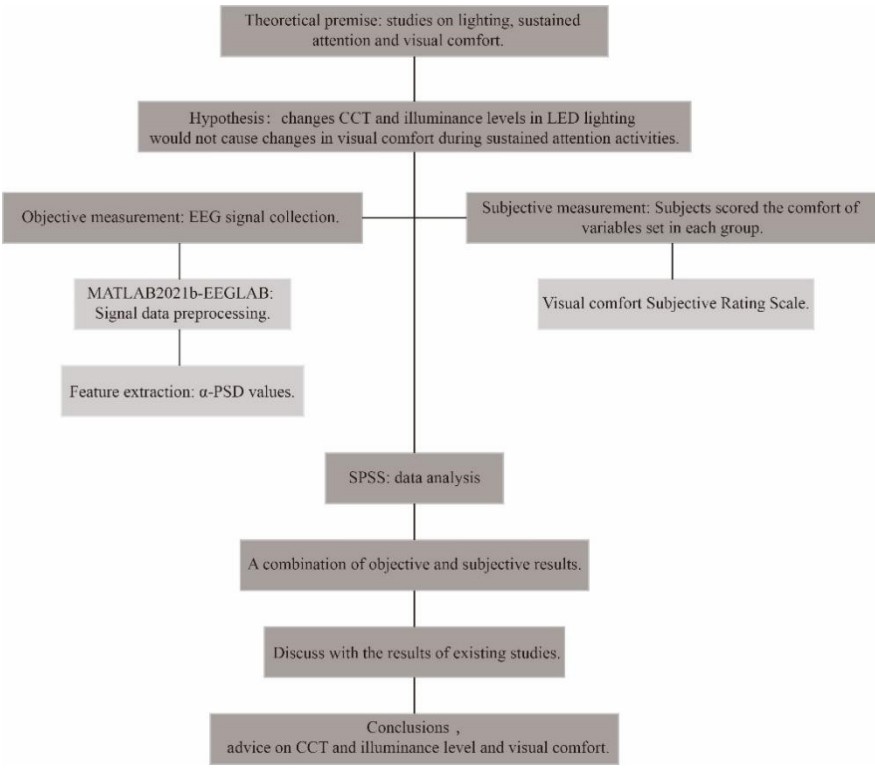

**Figure 1.** The theoretical framework.

In this study, the combination of typical CCTs and illuminance levels under different lighting conditions was selected as the variables. Next, the SART model was used. At the same time, the subjects' electrical EEG signals were collected by EEG equipment. After that, the EEGLAB toolbox running in MATLAB202lb was used to preprocess the EEG signals collected. Finally, we calculated the average power spectral density (PSD) of $\alpha$ waves and analyzed the data in SPSS to explore the effect of CCT and illuminance level of LED lighting on visual comfort during sustained attention activities. In addition, we asked the subjects to give a subjective score on the comfort level of each group of lights and conducted a conversation centered on the comfort level of CCT and illuminance, so as

to help us have a more comprehensive understanding of the subjects' subjective feelings during the experiment.

### 2.1. Experimental Subjects

To answer the experiment's aim, we selected 46 physically and mentally healthy university students (23 male, 23 female) as the experimental subjects. Subjects were 22–26 years of age (mean, 24.2 years). All subjects had normal vision and were right-handed as assessed by the Edinburgh Handedness Inventory as published by Oldfield in 1971 [21]. Ishihara plate test was used to test both achromatic and chromatic visual acuity for all subjects [22], and the test results of 46 subjects all met the test standards. No subjects had any psychiatric or neurological disorders, head trauma, history of smoking, alcohol addiction, or drug abuse. At 2 days before the experiment, the subjects had at least 8 h of sleep per day. No drugs were administered, except for a conductive gel to control skin resistance, and subjects were forbidden to take any drugs that might affect brain excitability and therefore visual comfort.

### 2.2. Apparatus

LED lighting equipment was used to simulate 9 groups of illumination conditions with different CCT and illuminance levels, assigned to the subjects in random order. According to the architectural lighting design standard (GB50034-2013) [23], when performing attention-related activities, the CCT of lighting conditions is classified into three groups: less than or equal to 3300 K as warm color; greater than 3300 K and less than 5300 K as neutral color; and greater than or equal to 5300 K as cool color. Commonly used colors include 3300 K (warm), 4300 K (neutral), and 5300 K (cool). The illuminance level could be roughly divided into three types, namely 300, 500, and 750 lx. Therefore, the CCT and illuminance levels of the lighting conditions in this study were set as 9 modes: 3300 K/300, 3300 K/500, 3300 K/750, 4300 K/300, 4300 K/500, 4300 K/750, 5300 K/300, 5300 K/500, 5300 K/750 lx.

We used the lamps provided by Tian Wen, a lamp manufacturer, which meet the National AA standard [24]. The adjustment range of CCT was 3000~5800 K, and the adjustment range of illumination level was 0~1000 lx. Before each experiment, we set and measured the CCT and illumination level of the lamps in the experiment by referring to performance requirements for table lamps for paper task [25], so as to ensure that the relevant parameters meet the requirements. The selection of measurement method and position was as follows: after the light source of the lamp worked steadily and normally above the work plane, the vertical projection point of the geometric center of the luminaire outlet was taken as the center of the circle, located directly in front of the eye. Within the projection range of the luminaire near the eye, the radius distance from the center of the circle was one third of 500 mm, and the CCT and illumination level were measured on the radius line at an interval of 30°. The average value was calculated after the sum of each point position as the experimental parameters (as shown in the Figure 2), and the measuring equipment used in the experiment was Lux Seeker, whose CCT measurement error was ±5%; illumination measurement error was ±5%.

An Active Two 64-lead EEG (BIOSEMI, Netherlands) was used in the experiment. The electrodes were mounted into the elastic cap according to the BIOSEMI position system. Reference electrodes were placed on the left and right ear mastoid processes, with CMS and DRL as grounding electrodes. We set the sampling rate to 1024 Hz in the built-in ActiView collection software. We used a conductive gel to control skin impedance below 5 k$\Omega$ and paid attention to the prompts in the software for electrode connections. The experimental environment is shown in Figure 3.

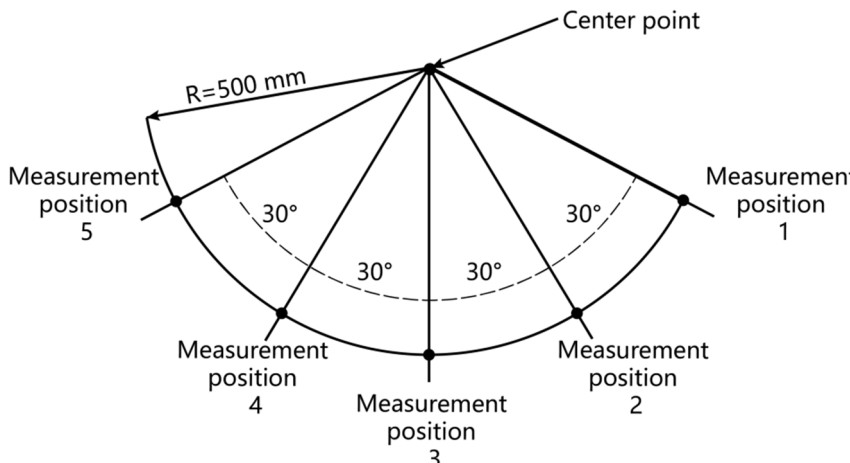

**Figure 2.** The selection of measurement method and position.

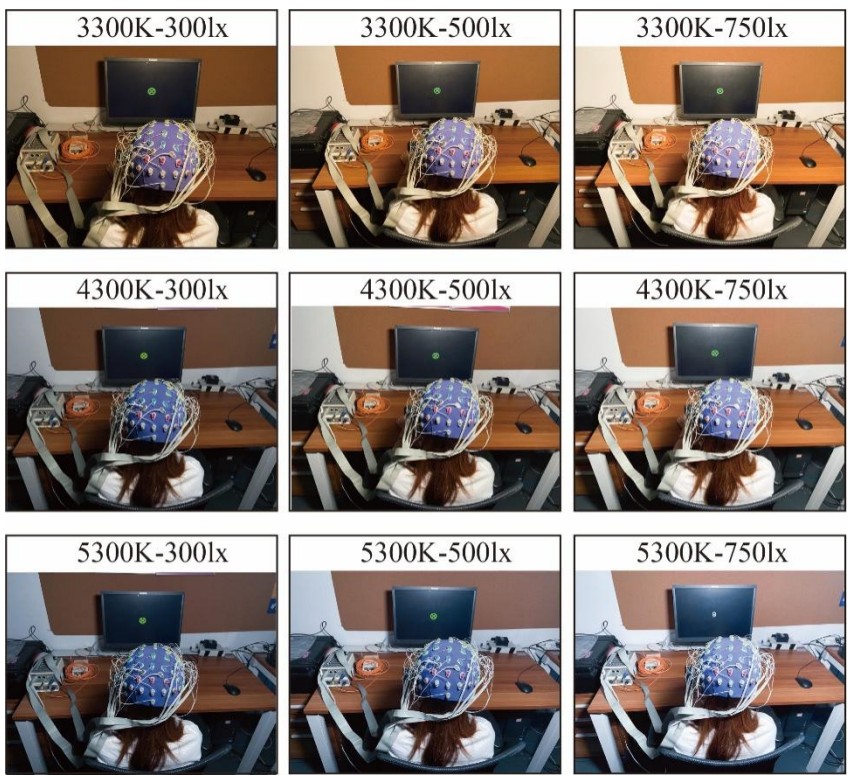

**Figure 3.** Schematic diagram of experimental environment.

Light acquisition by human eyes comes from many aspects. Visible light within a certain range may have a great influence on visual comfort. In order to effectively design lighting, some standards suggested appropriate illumination, color, distribution and type to improve visual comfort and enable people to perform visual tasks efficiently [26,27]. This means that lighting conditions in the room can affect the person's visual comfort and thus the perception of the test tasks. In addition, to ensure that the control variables provided the same ambient conditions of comfort for all subjects, the parameters of the monitor in the experiment were all the same.

### 2.3. Experimental Procedure

2.3.1. Sustained Attention to Response Test

Although the light on the screen was constant, changes in background lighting conditions can affect attention, mood, motivation, and task performance [28]. That meant that although the LED lighting we set would not affect the SART task on the monitor, it would affect the performance of the human eyes and thus affect the task outcome [29]. When conducting experiments in various lighting modes, the environmental conditions were set at 25 °C, 45% humidity, and a noise level of 24 dB. During the experiments, we made sure that other factors in the environment remained unchanged. This means that the subjects' test tasks were only affected by the lighting conditions in the room, so that the results were highly correlated with lighting conditions. The SART test was performed by wearing an EEG data-collection device and sitting in front of an HP screen with a resolution of 1920 × 1080 pixels. In the SART, the Go/No-go task is particularly useful for assessing sustained attention [30]. Participants sat in adjustable height seats facing the monitor. With a visual distance of 60 cm from the monitor, the subjects could adjust the height and horizontal distance of the sitting position. Adjustments of the workstation including display height, working distance, and angle were made in accordance with EN ISO 9241–5:1998 [31]. The Go/No-go task was used in our experiment to measure attention levels (i.e., "respond to numbers 1–9 with one exception: do not respond to number 3") (as shown in Figure 4).

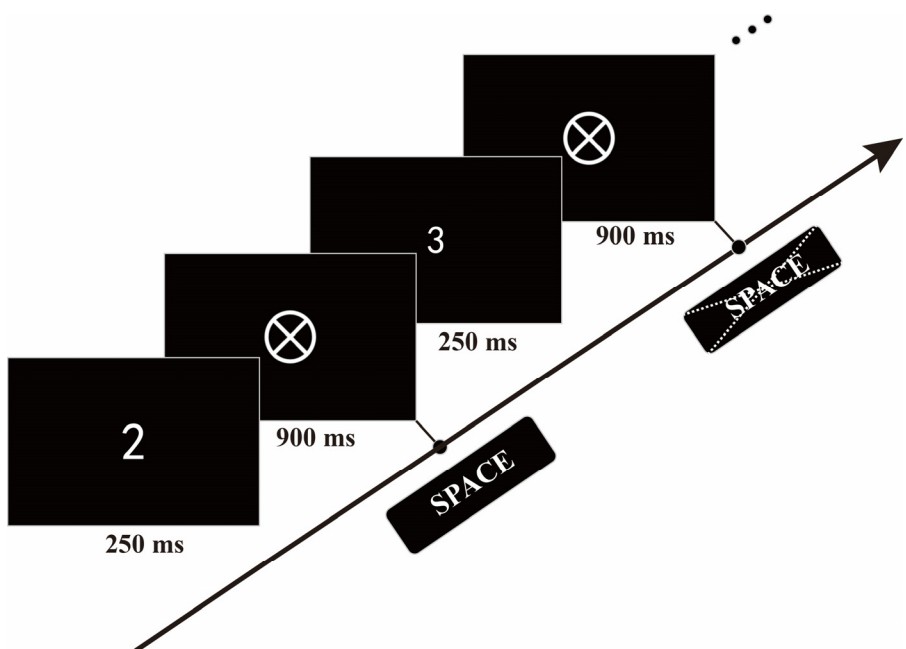

**Figure 4.** Sustained Attention to Response Task (SART).

The subjects' attention levels were assessed using 3 parameters of the SART: (1) the success of the No-go task: refused to respond to question number 3; (2) omission of Go tasks: did not respond to questions 1–9 other than number 3; (3) mean reaction time: calculated the mean and standard deviation of the time the subjects took to respond correctly to all questions except number 3.

In the experiment, a number (1–9) was displayed on the monitor in one of five randomly selected font sizes (48 points, 72 points, 94 points, 100 points, and 120 points). Each digit appeared for a short amount of time (250 ms), followed by masking for 900 ms. The duration of the experiment for the numbers and masking was 1.15 s. Before each set of the real experiment, a set of training trials was used to help the subjects enter a state of attention. Each digit 1–9 appeared exactly twice in the training trials. For each real experimental set, each digit appeared 25 times (225 times total), while each font size was used 45 times. One

set of experiments took about 4.3 min to complete. The participants were instructed to press the space bar in response to all stimuli except for question number 3. Each subject performed the SART under the above 9 different combinations of lighting environments. Each lighting experiment lasted about 11.3 min. The SART experiment took about 4.3 min. The subjects were required to rest their eyes for 2 min between tests to prevent fatigue and to facilitate the next set of experiments. In addition, before the SART experiment, we asked the subjects to perform visual adaptation under the lighting conditions of this group, and the adaptation time was about 5 min. Ultimately, 46 groups of data from the subjects were available for each combination of CCT and illuminance level.

We set a threshold for the validity of SART results, allowing for a maximum omission rate of 0.05% and a minimum No-go success rate of 60%. The participants were asked to remain focused throughout the experiment; otherwise, they were required to repeat the experiment.

### 2.3.2. Subjective Testing

After each group of SART experiments, we asked the subjects to rate the comfort of LED lighting. The evaluation scale refers to five indexes of tolerance scale [32]. Based on this, we made a slight change, using negative values to indicate unacceptable. The scale is: intolerable ($-4$), very uncomfortable ($-3$), uncomfortable ($-2$), slightly uncomfortable ($-1$), comfortable (0), keeping to one decimal place. This would make the subjects feel very intuitive but also conducive to our statistical analysis of data. In addition, the subject was asked to describe their feelings about the current LED lighting, such as fatigue, sleepiness, color perception, whether it affects SART, etc. (the evaluation scale is shown in Figure 5).

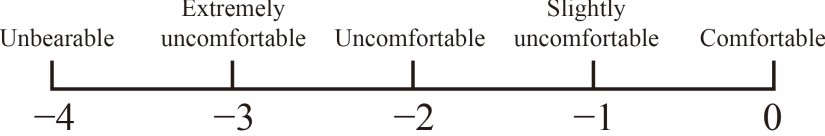

**Figure 5.** Evaluation scale.

### *2.4. Data Processing*
### 2.4.1. Data Pre-Processing

For the collected original EEG signals, the EEGLAB [33] toolbox was used to preprocess the data. The specific steps were as follows: First, the error test (wave amplitude changes too much) was deleted, and we detected whether there was bad conductance. Second, the sampling frequency was reduced to 256 Hz. Then, the bilateral mastoid electrodes were selected for re-reference. We used the *pop_eegfiltnew* function filtering tool in EEGLAB to filter the signal. The signal was then attenuated by a high-pass filter at 0.5 Hz to the EEG frequency artifacts (such as the artifacts generated by blinking and motion), the grid interference was eliminated by a notch filter at 50 Hz, and the $\alpha$ waves EEG data were extracted by a band-pass filter at 8–12 Hz and were divided into 900 ms epochs (100 ms before and 800 ms after stimulation). Baseline correction was performed. Finally, infomax ICA was used to remove eye movement artifacts and EMG artifacts. In addition, the data sections were inspected and screened manually, and the noise components were removed manually.

### 2.4.2. Feature Extraction

PSD represents the power of a signal with frequency and shows the energy intensity as a function of frequency. It is one of the most commonly used feature extraction methods in EEG research [34]. The method is based on frequency domain analysis to convert data from the time domain to the frequency domain. This conversion is based on the fast Fourier transform (FFT) to measure the discrete transform of the Fourier series and its inverse transform. Using this mathematical method of frequency analysis of complex waveform,

the EEG signal can be quantitatively analyzed [35,36]. Therefore, in this study, $\alpha$ waves PSD level was used as a quantitative feature to analyze the level of visual comfort.

## 3. Results

### 3.1. Analysis of PSD

We performed the analysis using repeated measures of variance (ANOVA), which included two subjects' internal factors labeled as illuminance level and CCT. We performed within-subject effect tests on the subjects' mean PSD to evaluate the effect of different CCT and illuminance levels on visual comfort. The null hypothesis was: CCT and illumination level had no effect on the mean PSD. We found significant differences in both CCT ($p = 0.006$) and illuminance level ($p = 0.027$). At the same time, we tested the interaction effect of CCT and illuminance level ($p = 0.645$), and there was no significant difference. This indicated that there was no interaction effect between CCT and illuminance level on visual comfort (as shown in Tables 1 and 2). By studying the mean value of the $\alpha$-band PSD of the subjects during the SART task stage under different CCT and illuminance levels, we found that the mean PSD was the lowest when the CCT was 5300 K and the illuminance level was 500 lx, i.e., the subject's visual comfort was the lowest. When the CCT was 3300 K and the illuminance level was 300 lx, the mean PSD was the highest, i.e., the subject's visual comfort level was the highest (as shown in Figure 6).

**Table 1.** Mean value of PSD in various CCT and illuminance levels.

| CCT (K) | IL (lx) | Mean-P | SD |
|---|---|---|---|
| | 300 | −44.62 | 6.17 |
| 3300 | 500 | −46.88 | 5.68 |
| | 750 | −46.17 | 5.58 |
| | 300 | −45.34 | 6.85 |
| 4300 | 500 | −48.32 | 6.44 |
| | 750 | −47.05 | 5.22 |
| | 300 | −46.28 | 5.87 |
| 5300 | 500 | −49.14 | 5.12 |
| | 750 | −47.52 | 4.98 |

IL stands for illumination level; Mean-P is the mean values of PSD.

**Table 2.** Test statistic result.

| | CCT | IL | CCT $\times$ IL |
|---|---|---|---|
| F | 7.09 | 9.22 | 0.504 |
| *p* value | 0.006 ** | 0.027 * | 0.645 |
| Partial $\eta^2$ | 0.490 | 0.515 | 0.278 |

** $p < 0.01$, * $p < 0.05$; IL stands for illumination level.

The paired comparisons were Bonferroni corrected in Table 3. Separate pairwise comparisons of CCT and illuminance levels revealed significant differences between CCT between 3300 and 5300 K ($p = 0.001$) and between 4300 and 5300 K ($p = 0.037$), while the differences between 3300 and 4300 K ($p = 0.279$) were insignificant. This indicated that the subjects' visual comfort level was lowest when the CCT was 5300 K. There were significant differences between the 300 and 500 lx ($p = 0.002$) and between 300 and 750 lx ($p = 0.018$) illuminance levels but no significant difference between 500 and 750 lx ($p = 0.289$). This showed that the subjects' visual comfort levels were lower when the illuminance level was 500 versus 300 lx and the subjects' visual comfort levels were lower when the illuminance level was 750 lx versus 300 lx (as shown in Table 3).

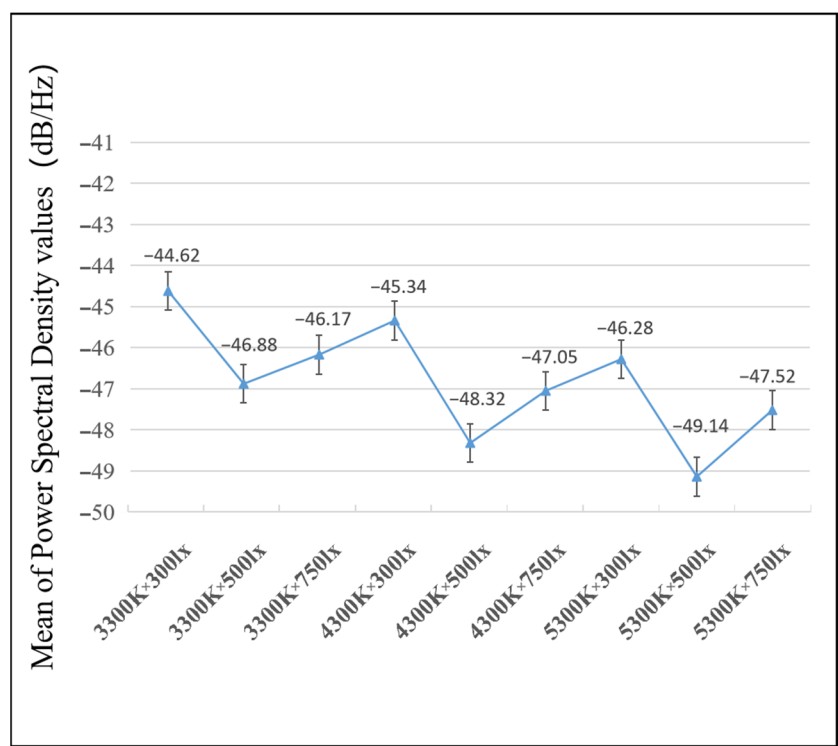

**Figure 6.** Different PSD means according to CCTs and illumination levels. The vertical line represents the standard error.

**Table 3.** Bonferroni-corrected pairwise comparison result.

| CCT (K) | | *p* Value | IL (lx) | | *p* Value |
|---|---|---|---|---|---|
| 3300 | 4300 | 0.279 | 300 | 500 | 0.002 ** |
| 3300 | 5300 | 0.001 ** | 300 | 750 | 0.018 * |
| 4300 | 5300 | 0.037 * | 500 | 750 | 0.289 |

** $p < 0.01$, * $p < 0.05$; IL stands for illuminance level.

### 3.2. Evaluation of SART Parameters

We performed repeated measures analysis of variance (ANOVA), including two within-subject factors labeled as illuminance level and CCT, tested for within-subject effects on the mean number of errors in the No-go tasks, and calculated the mean response time and its standard deviation. The null hypothesis was: CCT and illumination level had no effect on the mean number of errors in the No-go tasks. When CCT ($p = 0.005$) was 5300 K and illuminance level ($p = 0.011$) was 500 lx for No-go tasks, the mean number of errors was the lowest, with an average of 7.67. When the CCT was 3300 K and the illuminance level was 300 lx, the mean value of the number of errors was 20.25. In addition, because the omission rate of GO tasks was lower than 0.05%, it was not statistically significant; thus, it was not included in the statistics. The mean response time was not significant in the correlation between CCT ($p = 0.598$) and illuminance level ($p = 0.336$). This indicated that CCT and illuminance level have no significant effect on the average response time of subjects to SART. In addition, in order to better compare scale score and No-go task values across different lighting conditions, we changed the score of comfort scale into positive numbers, and we used positive numbers for the scale on the primary *y*-axis with No-go task values on the secondary *y*-axis. (as shown in Figure 7, Tables 4–6).

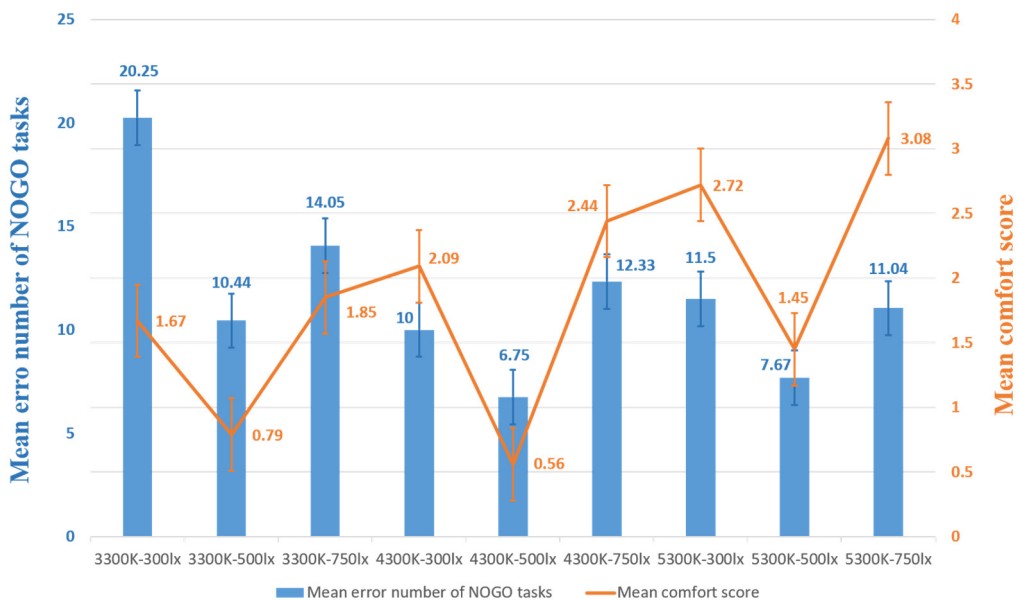

**Figure 7.** Different mean error numbers of NOGO tasks according to CCTs and illumination levels and mean comfort score. The vertical line represents the standard error.

**Table 4.** Mean error number of No-go tasks.

| CCT (K) | IL (lx) | Mean-N | SD |
|---|---|---|---|
| 3300 | 300 | 20.25 | 7.88 |
| | 500 | 10.44 | 5.29 |
| | 750 | 14.05 | 5.66 |
| 4300 | 300 | 10.00 | 9.24 |
| | 500 | 6.75 | 8.45 |
| | 750 | 12.33 | 9.35 |
| 5300 | 300 | 11.50 | 7.98 |
| | 500 | 7.67 | 4.60 |
| | 750 | 11.04 | 5.47 |

IL stands for illumination level; Mean-N is the mean error number of No-go tasks.

**Table 5.** Test statistic result.

| | CCT | IL | CCT $\times$ IL |
|---|---|---|---|
| F | 7.09 | 3.22 | 2.67 |
| *p* value | 0.005 ** | 0.011 * | 0.689 |
| Partial $\eta^2$ | 0.477 | 0.298 | 0.033 |

** $p < 0.01$, * $p < 0.05$; IL stands for illumination level.

**Table 6.** Bonferroni-corrected pairwise comparison result.

| CCT (K) | | *p* Value | IL (lx) | | *p* Value |
|---|---|---|---|---|---|
| 3300 | 4300 | 0.058 | 300 | 500 | 0.026 * |
| 3300 | 5300 | 0.025 * | 300 | 750 | 0.166 |
| 4300 | 5300 | 0.179 | 500 | 750 | 0.021 * |

* $p < 0.05$; IL stands for illumination level.

*3.3. Grey Relation Analysis*

We carried out non-dimensional processing (averaging) on the data, solved the grey correlation value between the contrast sequence and the feature sequence, and then solved the grey correlation degree and sorted the grey correlation degree to reach a conclusion. Grey correlational degree analysis was carried out for two evaluation items (CCT, illuminance level), and the mean value of PSD, the mean error number of NOGO tasks and the mean comfort scores were taken as the "reference value". When we used the grey correlation analysis, the distinguishing coefficient was set at 0.5. Then, we calculated the correlation degree as a basis for evaluation. The correlation degree ranges from 0 to 1, and the larger the value is, the stronger the relation is with the "reference value".

For the two evaluation items with the mean value of PSD as "reference value", CCT was rated the highest (correlation degree value: 0.678), followed by illuminance level (correlation degree value: 0.527). For the two evaluation items with the mean error number of NOGO tasks as "reference value", CCT was rated the highest (correlation degree value: 0.618), followed by illuminance level (correlation degree value: 0.592), and for the two evaluation items with the mean comfort scores as "reference value", CCT was rated the highest (correlation degree value: 0.636), followed by illuminance level (correlation degree value: 0.534). Therefore, we believe that CCT is more correlated with visual comfort in sustained attention tasks than illuminance level.

## 4. Discussion

In terms of the effects of CCT and illuminance levels on perceived visual comfort, participants indicated that different levels of visual comfort were associated with changes in CCT and illuminance levels. Similar findings were found in some field lighting simulations [37,38]. For example, Yang and Jin et al. [37] also observed that at 650 and 1000 lx illuminance levels, users expressed different levels of visual comfort. In our study, the mean PSD of alpha waves had a significant effect, depending on the LED lighting parameters during the SART. The highest mean PSD of $\alpha$ waves appeared when the CCT was at 3300 K and the illuminance level at 300 lx, which indicated that this group of LED lighting conditions brings the most comfortable visual experience. The findings of this study suggest that the CCT of 3300 K was the ideal CCT for the comfort level of the subjects during prolonged concentration activities, despite the fact that this result differed somewhat from the subjective comfort score. The pre-stimulus alpha waves power was shown to be decreased under lighting circumstances with higher CCT and illuminance levels in Yang et al.'s study [37]. In our study, the highest mean number of errors in No-go tasks occurred when the CCT was 3300 K and the illuminance level was 300 lx, suggesting that the lighting environment with high visual comfort would not always result in the enhancement of task performance. Low CCT and illuminance level can also lead to decreased clarity, distraction, fatigue and other results [39,40], which was obviously adverse to the improvement of tasks performance. Additionally, 4000 K was one of the most comfortable CCTs in this study's participants' subjective evaluation scores at all illuminance levels, and the error rate of No-go tasks was low at this CCT, with the lowest error rate at 500 lx. This finding was also consistent with several lighting results [4,35,41]. For example, Sivaji et al. [42] found that at 400 lx, office users rated 4000 K as more visually comfortable than 2700 and 6200 K. Similarly, Shamsul et al. [4] observed that 4000 K was considered the most comfortable by participants at the same illuminance level among the CCT levels of 3000, 4000, and 6500 K. In Baniya et al.'s [41] field lighting simulation, a combination of 4000 K and 750 lx was determined to be the most visually comfortable lighting condition.

Although we conducted an interaction effect test on CCT and illuminance level, trying to find the relationship between CCT and illuminance level, there was no significant effect. This result was consistent with the research results of Shen et al. [43], who found that the influence of CCT and illuminance level was independent from each other in the multivariate analysis of subjective evaluation of indoor lighting environment. Other factors may also account for this result, such as small number of samples of subjects in this experiment,

fatigue and alertness caused by the long experiment time and the limited division of CCT and illuminance level.

## 5. Conclusions

In this study, we combined CCT and illuminance level parameters for daily LED lighting. Subjects performed the SART task in nine different combinations of lighting conditions. Objectively, we analyzed the EEG data of the subjects during the experiment. Subjectively, we analyzed the subjects' subjective visual comfort scale. The results showed that this was a rigorous and accurate research method. We found that when people perform sustained attention activities, lower CCT and lower illuminance level will bring us better visual comfort, and the results showed that the LED lighting with CCT of 3300 K and illuminance of 300 lx was more comfortable than other combined conditions. In the subjective perception of subjects, 4000 K CCT also had good visual comfort evaluation and caused good task performance. In this study, LED lighting parameters had no significant effect on the response time of SART task and the omission rate of Go task.

The results were limited to specific measurements and tools, and the use of multiple methods and tools to examine the relationship between CCT and illuminance levels and visual comfort may enhance the validity of the study. The focus of this experiment was to study the correlation between lighting parameters and sustained attention from the ergonomic perspective, so as to obtain the most helpful lighting parameters for sustained attention works and to provide reference for lighting tool design. However, due to the differences in luminescence principle, different color rendering, and different materials, the same color temperature of its chromatography may vary, which involves the professional knowledge of lighting tool design. Future studies will explore the effects of different spectral distribution of the LED light sources on sustained attention. At the same time, the influence of other parameters of lighting conditions, such as background lighting and ambient light, on lighting parameters in this study was not considered. In our study, the LED lighting values were measured at fixed points in an enclosed environment. In an actual building scene, the daylight entering the building will not only change with the passage of time, but will also be affected by the location, weather and other factors, which is difficult to control [12]. Therefore, this experiment adopts indoor LED lighting to simulate the real lighting environment. However, we have to consider the presence of sunlight or illumination from other sources in the light environment of the actual scene, which leads to higher CCT and illuminance level values. Further study of the potential effect of other parameters in lighting conditions on perceptual efficacy will be one of the topics of our future research.

**Author Contributions:** Conceptualization, X.F. and X.J.; methodology, X.F.; software, X.J.; validation, X.F., D.F. and X.J.; formal analysis, X.J. and T.W.; investigation, X.J. and T.W. All authors have read and agreed to the published version of the manuscript.

**Funding:** This research was funded by the research and development of intelligent learning environment regulation system, grant number SKY-HX-20210233.

**Institutional Review Board Statement:** The study was conducted in accordance with the Declaration of Helsinki and was approved by the Institutional Review Board of Institute of Industrial Design of Zhejiang University of Technology (SKY-HX-20210233, 23 November 2021).

**Informed Consent Statement:** Informed consent was obtained from all subjects involved in the study. Ethical committee: School of Design and Architecture, Zhejiang University of Technology. Research number: SKY-HX-20210233.

**Conflicts of Interest:** The authors declare no conflict of interest.

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
