# Peer review of "The Effect of Correlated Color Temperature and Illumination Level of LED Lighting on Visual Comfort during Sustained Attention Activities"

_sustainability, doi:10.3390/su15043826_

Round 1
Reviewer 1 Report
The work in its basic idea is a repetition of Kruithof's well-known research. The difference lies primarily in the LED light sources used in the experiment instead of fluorescent lamps. From this point of view, the work does not bring anything new apart from the analysis of objective results based on EEG signals.
I notice the following faults of the work, which should be corrected before the editors decide to publish . The article lacks details regarding the description of the stations where the tests were conducted. I mean the size of the monitor (viewing angle) and the distance of the test person from the monitor.
2. There is no information on which surface the lighting level was controlled on (on the desk top?, on the floor?, on the wall?, on the keyboard?)
3. There is no information about the spectral distribution of the LED light source. The CCT color temperature alone does not provide unambiguous information about the spectral distribution of the illuminating light
4. On what premises is the research hypothesis based that CTT and E (lx) in the vicinity of the monitor may affect the comfort of performing test tasks. Changing the lighting conditions in the room (E(lx) and CCT (K) in the vicinity of the monitor do not affect the perception of test tasks perceived on the monitor. The computer monitor autonomously generates test marks and their background (L, CCT) on the screen by independent of the lighting in the room own luminance and color of characters.
Author Response
Thank you very much for your comments about our paper.
After carefully studying your advice, we have updated the manuscript and revised it accordingly. We have also changed some of the statements and descriptions in the article.
The detailed corrections are listed below point by point:
Reviewer 1’ Major Comment No. 1:
The work in its basic idea is a repetition of Kruithof's well-known research. The difference lies primarily in the LED light sources used in the experiment instead of fluorescent lamps. From this point of view, the work does not bring anything new apart from the analysis of objective results based on EEG signals.
Response:
Thanks for your comments, and the innovation of this paper lies in:
- The innovation of the experimental purpose: the table lamp is a necessary for modern office workers. Previous studies focused on the relationship between single lighting parameters and vision, such as the effects of myopia and comfort. There are few studies on the correlation between lighting parameters and visual comfort in sustained attention activities. Kruithof's research focused on the impact of fluorescent lamps on user comfort. In contrast, our experiment was more in line with the current practical scenario application.
- Innovation of experimental methods: in this experiment, we explored the relationship between main parameters of LED lights (color temperature and illumination) and the visual comfort in sustained attention activities. On the other hand, this study further analyzed the relationship between comfort and sustainable attention, and added EEG signal testing to objectively verify the lighting parameters most consistent with sustained attention. In addition, we gave the design suggestions of lighting tools and the parameter adjustment suggestions under the user's actual scene.
Reviewer 1’ Major Comment No. 2:
I notice the following faults of the work, which should be corrected before the editors decide to publish. The article lacks details regarding the description of the stations where the tests were conducted. I mean the size of the monitor (viewing angle) and the distance of the test person from the monitor.
Response:
We described the conditions of the experimental scene in more detail.
The revised text is as follows:
Participants sat in adjustable height seats facing the monitor. With a visual distance of 60 centimeters from the monitor, the subjects could adjust the height and horizontal distance of the sitting position. Adjustments of the workstation including display height, working distance, angle, were made in accordance with EN ISO 9241–5:1998[1].
[1] ISO I S O. 8995-1: 2002 (CIE S 008/E: 2001) Lighting of Indoor Work Places. ISO Organización Internacional de Normalización, 2002.
Reviewer 1’ Major Comment No. 3:
There is no information on which surface the lighting level was controlled on (on the desk top?, on the floor?, on the wall?, on the keyboard?)
Response:
We provided supplementary instructions on the measurement settings of lights.
The revised text is as follows:
We used the lamps provided by Tian Wen, a lamp manufacturer. And the lamps meet the National AA standard [1]. The adjustment range of CCT was 3000 K ~ 5800 K, and the adjustment range of illumination level was 0 ~ 1000 lx. Before each experiment, we set and measured the CCT and illumination level of the lamps in the experiment by referring to performance requirements for table lamps for paper task [2], so as to ensure that the relevant parameters meet the requirements. The selection of measurement method and position was as follows: after the light source of the lamp worked steadily and normally above the work plane, took the vertical projection point of the geometric center of the luminaire outlet as the center of the circle, located directly in front of the eye. Within the projection range of the luminaire near the eye, the radius distance from the center of the circle was one third of 500 mm, and the CCT and illumination level were measured on the radius line at an interval of 30°. And calculated the average value after the sum of each point position as the experimental parameters (as shown in the Figure 1). And, the measuring equipment used in the experiment was Lux Seeker, whose CCT measurement error was ± 5%, illumination measurement error was ± 5%.
Figure 1 The selection of measurement method and position.
[1] Standard for lighting design of buildings. National Standards GB/T 50034-2013.
[2] Performance requirements for table lamps for paper task. National Standards GB/T 9473-2017
Reviewer 1’ Major Comment No. 4:
There is no information about the spectral distribution of the LED light source. The CCT color temperature alone does not provide unambiguous information about the spectral distribution of the illuminating light.
Response:
Thanks for pointing out the shortcomings of this article. We added a discussion of this issue to our article.
The revised text is as follows:
And, the focus of this experiment was to study the correlation between lighting parameters and sustained attention from the ergonomic perspective, so as to obtain the most helpful lighting parameters for sustained attention works and provide reference for lighting tool design. However, due to the differences in luminescence principle, different color rendering, different materials, the same color temperature of its chromatography may vary, which involves the professional knowledge of lighting tool design. Future studies will explore the effects of different spectral distribution of the LED light sources on sustained attention.
Reviewer 1’ Major Comment No. 5:
On what premises is the research hypothesis based that CTT and E (lx) in the vicinity of the monitor may affect the comfort of performing test tasks. Changing the lighting conditions in the room (E(lx) and CCT (K) in the vicinity of the monitor do not affect the perception of test tasks perceived on the monitor. The computer monitor autonomously generates test marks and their background (L, CCT) on the screen by independent of the lighting in the room own luminance and color of characters.
Response:
Thank you for your question, we added the answer to this question in the article.
The revised text is as follows:
Light acquisition by human eyes comes from many aspects. Visible light within a certain range may have a great influence on visual comfort. In order to effectively design lighting, some standards suggested appropriate illumination, color, distribution and type to improve visual comfort and enable people to perform visual tasks efficiently [1-2]. This means that lighting conditions in the room can affect the subjects' visual comfort and thus the perception of the test tasks. In addition, to ensure that the control variables provided the same ambient conditions of comfort for all subjects, the parameters of the monitor in the experiment were all the same.
[1] Phillips D. Lighting modern buildings. Routledge, 2013.
[2] Boyce P, Raynham P. SLL lighting handbook. Cibse, 2009.

Reviewer 2 Report
Your study raises important and relevant issues. However, it reads more like a report than an academic article, with questionable assertions and no detailed citations or arguments. The study lacks a theoretical framework or paradigm to provide context.
Author Response
Thank you very much for your comments about our paper.
After carefully studying your advice, we have updated the manuscript and revised it accordingly. We have also changed some of the statements and descriptions in the article.
Reviewer 2’ Major Comment No. 1:
Your study raises important and relevant issues. However, it reads more like a report than an academic article, with questionable assertions and no detailed citations or arguments. The study lacks a theoretical framework or paradigm to provide context.
Response:
Thanks for your suggestions. We revised the article according to your suggestions.
The revised text is as follows:
This was an experimental study, not a purely theoretical one. We explored the visual comfort caused by LED lighting for daily activities. The research method combined physiological signal measurement with subjective scale, which has been proved to be rigorous [1]. The theoretical framework of this paper is as follows:
In the study of related theories, this paper verified the theoretical feasibility of the thesis by citing two aspects. First, the paper cited the correlation between CCT, illuminance level and virtual comfort level to prove that there is a significant correlation between them. Second, the paper cited the correlation between visual comfort and sustained attention to prove that comfort level and sustained attention are strongly correlated. Next, we hypothesized that changes CCT and illuminance levels in LED lighting would not cause changes in visual comfort during sustained attention activities. Then, we selected the daily value range of CCT and illuminance level [2], graded them and combined them as experimental variables. Then, we collected the EEG signals of the subjects in the sustained attention to response test (SART) tasks and extracted the features of the frequency band related to visual comfort. At the same time, after each group of experiments, subjects would conduct subjective visual comfort evaluation [3]. Finally, we synthesized the results of objective physiological signals and the results of subjective scales as the basis for the conclusion of this paper. (as shown in Figure 1)
Figure 1 The theoretical framework.
[1] Fotios.; Kent M. Measuring discomfort from glare: recommendations for good practice. Leukos, 2021, 17(4): 338-358
[2] Method of measuring the color of light sources. National Standards GB/T 7922-2008.
[3] Speranza F.; Tam W J.; Renaud R. et al. Effect of disparity and motion on visual comfort of stereoscopic images. stereoscopic displays and virtual reality systems XIII. SPIE, 2006, 6055: 94-103.

Reviewer 3 Report
The entitled "The effect of correlated color temperature and illumination level of LED lighting on visual comfort during sustained attention activities" assessed various combination of LED lighting on human visual comfort. The experiment was well designed. However, there is no lines for the whole manuscript and that I could not give the detailed comments based on the status now. I would suggest the authors re-submit their manuscript by handling the issue of lines using word.
Author Response
Thanks for your suggestions. We added line numbers to the manuscript.

Reviewer 4 Report
The paper presents an interesting experiment that varied both illuminance and CCT. Although the results were generally anticipated, the authors also collected EEG measurements and offered design recommendations. Both were appreciated. I believe that the paper would benefit if more literature aligned to the topic were included, and parts of the method were better explained. I have documented both these remarks, in more detail, in the comments below. I hope that the authors find these useful when revisiting their work:
#1: Intro. For the representation of what is considered “good lighting”, I thought that it would be more apt to reference an established international lighting standard (e.g., CIE S 008/E-2001. Lighting for indoor workplaces, or EN 12463-1, 2021. Light and Lighting.)
#2: Intro: I believe that it is more appropriate to say that illuminance and CCT are among some of the most important characteristics to human cognitive activities. Other features are arguable just as important, such as: color rendition and glare. It may also be more accurate to refine this to human cognitive activities that involve visual performance, to be more specific to the context of this research.
#3: Intro: The literature on CCT and visual perception could be expanded. Studies have produced very similar influences for CCT at controlled illuminances from chromatic glazing applications (Liang, 2019: 2021), showing that visual acuity was higher under higher color temperatures, yet a lower CCT was still preferred by people. Given the vast interest on color lighting, I thought that the authors could review more studies:
- Liang 2019. Development of experimental methods for quantifying the human response to chromatic glazing
- Liang 2021. The effect of thermochromic windows on visual performance and sustained attention
#4: Intro: Although the statement about subjective comfort assessments is accurate, the reference [11] on airline passenger comfort was a rather odd inclusion to support this study, particularly when considering it didn’t seem to relate to either visual comfort or lighting. Fotios, 2021. Measuring discomfort from glare: recommendations for good practice, both raises the problems when using subjective scales for visual comfort research, and also advocates the measurement of objective measures to produce more rigorous conclusions.
#5: Intro: It would be helpful to readers to contextualize “sustained attention” to LED. For instance, did the authors envision their work helping students maintain their attention longer on illuminated papers or books?
#6: Intro: “Help users to” -> “This will help users to”
#7: S2.1: Please consider replacing “ensure the accuracy” -> “to answer the experiment’s aim”
#8: S2.1: Did the authors test both achromatic and chromatic visual acuity?
#9: S2.2: Please detail how and where light conditions were measured (e.g., participants’ eye or at the desk.)
#10: The Go/No-go task was well explained, but I believe that this was displayed on a self-illuminated computer monitor. If this is the case, the changes in the lighting condition (i.e. illuminance and CCT) only influenced the background conditions, and not the visual task used to measure sustained attention. It would be useful to explain how this may have influenced the outcome.
#11: S3.2.1: Since the task itself was not influenced by changes in the lighting conditions, it might be that the differences in sustained attention were likely brought on by alerting effects created by different illuminances and CCT. It would be helpful to include how long participants experienced each lighting condition, and if they were provided a period of visual adaptation before performing each task.
#12: S2.3.2: It was generally unclear why the comfort scale for evaluating visual comfort followed a reference [21] used for accessing the thermal environment. There are many scales used for evaluating both subjective brightness, color appearance, and other facets of visual comfort, which could have been considered.
#13: Results: Please include the effect sizes for the statistical comparisons. Moreover, were the paired comparisons corrected (e.g., Bonferroni) in Table 3?
#14: Figure 5. Although the authors explained why the scale took on negative numbers, its benefit was not particular clear in the analysis. Would it be possible to use positive numbers for the scale on the primary y-axis, and have the NoGo task values on the secondary y-axis? This would help compare these values across the different lighting conditions.
#15: It would be useful to briefly mention how these results may be influenced in buildings that are also daylit, and would have variable brightness and CCT.
Author Response
Thank you very much for your comments about our paper.
After carefully studying your advice, we have updated the manuscript and revised it accordingly. We have also changed some of the statements and descriptions in the article.
Reviewer 4’ Major Comment No. 1:
The paper presents an interesting experiment that varied both illuminance and CCT. Although the results were generally anticipated, the authors also collected EEG measurements and offered design recommendations. Both were appreciated. I believe that the paper would benefit if more literature aligned to the topic were included, and parts of the method were better explained. I have documented both these remarks, in more detail, in the comments below. I hope that the authors find these useful when revisiting their work:
#1: Intro. For the representation of what is considered “good lighting”, I thought that it would be more apt to reference an established international lighting standard (e.g., CIE S 008/E-2001. Lighting for indoor workplaces, or EN 12463-1, 2021. Light and Lighting.)
Response:
Thanks for your suggestion, we referred to "Lighting of Indoor Work Places" to revise the article.
The revised text is as follows:
A good lighting environment provides us with an environment in which visual tasks can be performed efficiently and accurately without causing visual fatigue and discomfort [1].
[1] ISO I S O. 8995-1: 2002 (CIE S 008/E: 2001) Lighting of Indoor Work Places. ISO Organización Internacional de Normalización, 2002.
Reviewer 4’ Major Comment No. 2:
#2: Intro: I believe that it is more appropriate to say that illuminance and CCT are among some of the most important characteristics to human cognitive activities. Other features are arguable just as important, such as: color rendition and glare. It may also be more accurate to refine this to human cognitive activities that involve visual performance, to be more specific to the context of this research.
Response:
Our previous description of the importance of CCT and illuminance level was ambiguous, so we adopted the revised description you suggested in the article.
The revised text is as follows:
There are many factors related to LED lighting that affect human cognitive activities, such as color rendition, glare, illuminance level, luminance uniformity, CCT, etc [1]. CCT and illuminance level are among some of the most important characteristics to human cognitive activities [2].
[1] Samani S A.; The influence of light on student’s learning performance in learning environments: A knowledge internalization perspective. Journal of World Academy of Science, Engineering and Technology, 2011, 81.
[2] Shamsul, B.; Sia, C.; Ng, Y.; Karmegan, K. Effects of light’s colour temperatures on visual comfort level, task performances, and alertness among students. Am. J. Public Health Res. 2013, 1, 159–165.
Reviewer 4’ Major Comment No. 3:
#3: Intro: The literature on CCT and visual perception could be expanded. Studies have produced very similar influences for CCT at controlled illuminances from chromatic glazing applications (Liang, 2019: 2021), showing that visual acuity was higher under higher color temperatures, yet a lower CCT was still preferred by people. Given the vast interest on color lighting, I thought that the authors could review more studies:
- Liang 2019. Development of experimental methods for quantifying the human response to chromatic glazing
- Liang 2021. The effect of thermochromic windows on visual performance and sustained attention.
Response:
Thanks for your suggestions. We reviewed more studies on visual perception and lighting.
The revised text is as follows:
Liang et al. [1-2] simulated the effects of CCTs and illuminance levels in different light conditions on visual performance and attention level of subjects as well as occupant comfort when passing through thermochromic windows. The results showed that visual acuity was higher under higher CCT, yet a lower CCT was still preferred by subjects and more natural and acceptable lighting conditions were found in bronze-tinted conditions.
[1] Liang R.; Kent M.; Wilson R. et al. Development of experimental methods for quantifying the human response to chromatic glazing. Building and Environment, 2019, 147: 199-210.
[2] Liang R.; Kent M.; Wilson R. et al. The effect of thermochromic windows on visual performance and sustained attention. Energy and Buildings, 2021, 236: 110778.
Reviewer 4’ Major Comment No. 4:
#4: Intro: Although the statement about subjective comfort assessments is accurate, the reference [11] on airline passenger comfort was a rather odd inclusion to support this study, particularly when considering it didn’t seem to relate to either visual comfort or lighting. Fotios, 2021. Measuring discomfort from glare: recommendations for good practice, both raises the problems when using subjective scales for visual comfort research, and also advocates the measurement of objective measures to produce more rigorous conclusions
Response:
Thanks for your suggestions. In view of the fact that only a small part of the content of the original reference [11] was related to our research, we deleted it and reviewed the new literature for supplementary explanation.
The revised text is as follows:
Using subjective scales to assess visual comfort while using objective measures to reach more accurate conclusions was a proven method [1].
[1] Fotios.; Kent M. Measuring discomfort from glare: recommendations for good practice. Leukos, 2021, 17(4): 338-358
Reviewer 4’ Major Comment No. 5:
#5: Intro: It would be helpful to readers to contextualize “sustained attention” to LED. For instance, did the authors envision their work helping students maintain their attention longer on illuminated papers or books?
Response:
We revised this article to make it easier for readers to understand "sustained attention" under LED.
The revised text is as follows:
The purpose of this study is to explore the relationship of EEG signals, SART, and subjective evaluation, to test the effects of different CCTs and illuminance levels on visual comfort when users use LED lighting for cognitive activities that require us to attention longer, and to determine the best combination of CCT and illuminance levels. For instance, when reading or writing with an LED table lamp with adjustable CCT and illuminance levels, we can adjust it to the right parameters to help maintain our attention for longer periods of time without causing visual discomfort.
Reviewer 4’ Major Comment No. 6:
#6: Intro: “Help users to” -> “This will help users to”
Response:
We corrected the expression according to your suggestion.
The revised text is as follows:
This will help users to achieve the best visual comfort while completing sustained attention tasks, so as to improve their satisfaction with work and study.
Reviewer 4’ Major Comment No. 7:
#7: S2.1: Please consider replacing “ensure the accuracy” -> “to answer the experiment’s aim”
Response:
We revised the description.
The revised text is as follows:
To answer the experiment’s aim, we selected 24 physically and mentally healthy university students (12 male, 12 female) as the experimental subjects.
Reviewer 4’ Major Comment No. 8:
#8: S2.1: Did the authors test both achromatic and chromatic visual acuity?
Response:
The subjects were tested both achromatic and chromatic visual acuity, but we overlooked the description in the article, so we made a supplement to it.
The revised text is as follows:
All subjects were tested both achromatic and chromatic visual acuity, and the test results of 24 subjects all met the test standards.
Reviewer 4’ Major Comment No. 9:
#9: S2.2: Please detail how and where light conditions were measured (e.g., participants’ eye or at the desk.)
Response:
We provided supplementary instructions on the measurement settings of lights.
The revised text is as follows:
Participants sat in adjustable height seats facing the monitor. With a visual distance of 60 centimeters from the monitor, the subjects could adjust the height and horizontal distance of the sitting position. Adjustments of the workstation including display height, working distance, angle, were made in accordance with EN ISO 9241–5:1998[1].
We used the lamps provided by Tian Wen, a lamp manufacturer. And the lamps meet the National AA standard [2]. The adjustment range of CCT was 3000 K ~ 5800 K, and the adjustment range of illumination level was 0 ~ 1000 lx. Before each experiment, we set and measured the CCT and illumination level of the lamps in the experiment by referring to performance requirements for table lamps for paper task [3], so as to ensure that the relevant parameters meet the requirements. The selection of measurement method and position was as follows: after the light source of the lamp worked steadily and normally above the work plane, took the vertical projection point of the geometric center of the luminaire outlet as the center of the circle, located directly in front of the eye. Within the projection range of the luminaire near the eye, the radius distance from the center of the circle was one third of 500 mm, and the CCT and illumination level were measured on the radius line at an interval of 30°. And calculated the average value after the sum of each point position as the experimental parameters (as shown in the Figure 1). And, the measuring equipment used in the experiment was Lux Seeker, whose CCT measurement error was ± 5%, illumination measurement error was ± 5%.
Figure 1 The selection of measurement method and position.
[1] ISO I S O. 8995-1: 2002 (CIE S 008/E: 2001) Lighting of Indoor Work Places. ISO Organización Internacional de Normalización, 2002.
[2] Standard for lighting design of buildings. National Standards GB/T 50034-2013.
[3] Performance requirements for table lamps for paper task. National Standards GB/T 9473-2017.
Reviewer 4’ Major Comment No. 10:
#10: The Go/No-go task was well explained, but I believe that this was displayed on a self-illuminated computer monitor. If this is the case, the changes in the lighting condition (i.e. illuminance and CCT) only influenced the background conditions, and not the visual task used to measure sustained attention. It would be useful to explain how this may have influenced the outcome.
Response:
Thank you for your question, we added an explanation of this effect to the paper.
The revised text is as follows:
Light acquisition by human eyes comes from many aspects. Visible light within a certain range may have a great influence on vision. Although the light on the screen was constant, changes in background lighting conditions can affect attention, mood, motivation, and task performance [1]. That meant that although the LED lighting we set would not affect the SART task on the monitor, it would affect the performance of the human eyes and thus affect the task outcome [2].
[1] Kralikova R.; Wessely E. LIGHTING QUALITY, PRODUCTIVITY AND HUMAN HEALTH. Annals of DAAAM & Proceedings, 2016, 27.
[2] Konstantzos I.; Sadeghi S A.; Kim M. et al. The effect of lighting environment on task performance in buildings–A review. Energy and Buildings, 2020, 226: 110394.
Reviewer 4’ Major Comment No. 11:
#11: S3.2.1: Since the task itself was not influenced by changes in the lighting conditions, it might be that the differences in sustained attention were likely brought on by alerting effects created by different illuminances and CCT. It would be helpful to include how long participants experienced each lighting condition, and if they were provided a period of visual adaptation before performing each task.
Response:
We added details about the relationship between the subject and the lighting during the experiment.
The revised text is as follows:
Each lighting experiment lasted about 11.3 minutes. The SART experiment took about 4.3 minutes, with 2 minutes of eye rest between the experiments. In addition, before the SART experiment, we asked the subjects to perform visual adaptation under the lighting conditions of this group, and the adaptation time was about 5 minutes.
Reviewer 4’ Major Comment No. 12:
#12: S2.3.2: It was generally unclear why the comfort scale for evaluating visual comfort followed a reference [21] used for accessing the thermal environment. There are many scales used for evaluating both subjective brightness, color appearance, and other facets of visual comfort, which could have been considered.
Response:
Thank you for your advice. That was really inappropriate for this study to cite the literature on thermal comfort. To avoid ambiguity, we replaced reference that was more relevant to visual comfort [1].
The revised text is as follows:
[1] Speranza F.; Tam W J.; Renaud R. et al. Effect of disparity and motion on visual comfort of stereoscopic images. stereoscopic displays and virtual reality systems XIII. SPIE, 2006, 6055: 94-103.
Reviewer 4’ Major Comment No. 13:
#13: Results: Please include the effect sizes for the statistical comparisons. Moreover, were the paired comparisons corrected (e.g., Bonferroni) in Table 3?
Response:
Thanks for your suggestions. We included the effect sizes for the statistical comparisons in Table 2 and 5. And, the paired comparisons were Bonferroni-corrected in Table 3 and 6.
The revised text is as follows:
Table 2. Results of intrasubjective effect test.
|
CCT |
IL |
CCT*IL |
|
F |
6.88 |
11.48 |
0.467 |
|
P |
0.008** |
0.012* |
0.759 |
|
Partial η2 |
0.461 |
0.590 |
0.129 |
**p<0.01, *p<0.05; IL stands for illumination level.
Table 5. Results of intrasubjective effect test.
|
CCT |
IL |
CCT*IL |
|
F |
6.56 |
4.51 |
0.30 |
|
P |
0.008** |
0.028* |
0.731 |
|
Partial η2 |
0.451 |
0.360 |
0.036 |
**p<0.01, *p<0.05; IL stands for illumination level.
The paired comparisons were Bonferroni-corrected in Table 3 and 6.
Table 3. Bonferroni-corrected pairwise comparison result.
CCT (K) |
P |
IL (lx) |
P |
||
3300 |
4300 |
0.387 |
300 |
500 |
0.005** |
3300 |
5300 |
0.002** |
300 |
750 |
0.021* |
4300 |
5300 |
0.049* |
500 |
750 |
0.400 |
**p<0.01, *p<0.05; IL stands for illuminance level.
Table 6. Bonferroni-corrected pairwise comparison result.
CCT (K) |
P |
IL (lx) |
P |
||
3300 |
4300 |
0.062 |
300 |
500 |
0.044* |
3300 |
5300 |
0.014* |
300 |
750 |
0.293 |
4300 |
5300 |
0.145 |
500 |
750 |
0.040* |
Reviewer 4’ Major Comment No. 14:
#14: Figure 5. Although the authors explained why the scale took on negative numbers, its benefit was not particular clear in the analysis. Would it be possible to use positive numbers for the scale on the primary y-axis, and have the NoGo task values on the secondary y-axis? This would help compare these values across the different lighting conditions.
Response:
We took your useful suggestion, changed the value of the scale to positive numbers, and modified the chart style.
The revised text is as follows:
In order to better compare scale score and NOGO task values across different lighting conditions, we changed the score of comfort scale into positive numbers. And, we used positive numbers for the scale on the primary y-axis, and have the NOGO task values on the secondary y-axis. ( as shown in Figure 6)
Figure 6. Different mean error number of NOGO tasks according to CCTs and illumination levels and mean comfort score; The vertical line represents the standard error.
Reviewer 4’ Major Comment No. 15:
#15: It would be useful to briefly mention how these results may be influenced in buildings that are also daylit, and would have variable brightness and CCT.
Response:
Thanks for your suggestions. We added a consideration to the question in the article.
The revised text is as follows:
In our study, the LED lighting values were measured at fixed points in an enclosed environment. While, in the actual building scene, the daylight entering the building will not only change with the passage of time, but also be affected by the location, weather and other factors, which is difficult to control [1]. Therefore, this experiment adopts indoor LED lighting to simulate the real lighting environment. However, we have to consider the presence of sunlight or illumination from other sources in the light environment of the actual scene, which leads to a higher CCT and illuminance level value in the conclusion than the actual one.
[1] Liang R.; Kent M.; Wilson R. et al. Development of experimental methods for quantifying the human response to chromatic glazing. Building and Environment, 2019, 147: 199-210.

Round 2
Reviewer 2 Report
The manuscript is well-written and revised in many issues that had been emerged and underlined in the previous version. I suggest its publication
Author Response
Thanks for your comments and very useful advice.
Reviewer 3 Report
This paper aims to explore the influence of correlated color temperature (CCT) and illuminance level in LED lighting parame ters on human visual comfort. The main problem lies in the limited number of samples as only 24 healthy college students (12 male and 12 female) were involved. Besides, more reliables statistical analysis method such as grey correlation matrix or structural equation model can be applied. I recommended the add of grey correlation matrix or structural equation model, more samples are also necessary if possible.
Author Response
Response:
Thanks for your valuable suggestions. Because of the COVID-19 pandemic, our research was limited in the number of subjects we could recruit. However, we continued to recruit and conduct experiments after submitting the paper. Up to now, 22 subjects have been added, totaling 46 subjects (23 males, 23 females, the ages ranged from 22 to 26 years old, average age was 24.2 years old). We have re-processed and analyzed the data of these subjects, and we have used the grey correlation matrix for further statistical analysis. The results showed that in sustained attention tasks, when the CCT was 3300 K or 4300 K and the illuminance level was 300 lx, the visual comfort of the subjects was better. And, the CCT was more correlated with visual comfort in sustained attention tasks than illuminance level. For the two evaluation items with the mean value of PSD as "reference value", CCT rated the highest (correlation degree value: 0.678), followed by illuminance level (correlation degree value: 0.527). For the two evaluation items with the mean error number of NOGO tasks as "reference value", CCT rated the highest (correlation degree value: 0.618), followed by illuminance level (correlation degree value: 0.592). And, for the two evaluation items with the mean comfort scores as "reference value", CCT rated the highest (correlation degree value: 0.636), followed by illuminance level (correlation degree value: 0.534).
The revised text is as follows:
3.3. Grey relation analysis
We carried out non-dimensional processing (averaging) on the data, solved the grey correlation value between the contrast sequence and the feature sequence, and then solved the grey correlation degree, and sorted the grey correlation degree to reach a conclusion. Grey correlational degree analysis was carried out for two evaluation items (CCT, illuminance level), and the mean value of PSD, the mean error number of NOGO tasks and the mean comfort scores were taken as the "reference value". When we used the grey correlation analysis, the distinguishing coefficient was set at 0.5. Then, we calculated the correlation degree as a basis for evaluation. The correlation degree ranges from 0 to 1, and the larger the value is, the stronger the relation is with the "reference value".
For the two evaluation items with the mean value of PSD as "reference value", CCT rated the highest (correlation degree value: 0.678), followed by illuminance level (correlation degree value: 0.527). For the two evaluation items with the mean error number of NOGO tasks as "reference value", CCT rated the highest (correlation degree value: 0.618), followed by illuminance level (correlation degree value: 0.592). And, for the two evaluation items with the mean comfort scores as "reference value", CCT rated the highest (correlation degree value: 0.636), followed by illuminance level (correlation degree value: 0.534). Therefore, we believe that CCT is more correlated with visual comfort in sustained attention tasks than illuminance level.

Reviewer 4 Report
Thank you considering all my previous comments. This was highly appreciated, and was also evident in the revised text included into the amended manuscript. I have a few more minor comments, which I believe could be considered to improve the clarify of the text, tables, and figures:
1. "was a proven" please consider changing to "was a recommended"
2. P4, L136: "All subjects were tested for both". Also please explain which tests were used for both (e.g. Landolt chart, or Ishihara plate test.)
3. Tables 2 and 5. Please include "Test statistic" as the title for the first column. Also "P" should be "p-value" This can also be applied to Tables 3 and 6.
4. Figure 6. Please provide a small space between the values and the upper confidence interval in the plot, so that they do not overlap with each other. Similarly, this can be applied to the text labels in Figure 7.
5. Please label to primary and secondary y-axes for Figure 7. The error bars could also be edited (e.g. by using different colors and/or thicknesses), so that readers can distinguish between which belongs to the bars and which belongs to the line for each lighting condition on the x-axis.
Author Response
Thank you very much for your comments
After carefully studying your advice, we have updated the manuscript and revised it accordingly.
The detailed corrections are listed below point by point:
Reviewer 4’ Major Comment No. 1:
Thank you considering all my previous comments. This was highly appreciated, and was also evident in the revised text included into the amended manuscript. I have a few more minor comments, which I believe could be considered to improve the clarify of the text, tables, and figures:
- "was a proven" please consider changing to "was a recommended"
Response:
Thanks for your suggestion, and we have revised it in the article.
The revised text is as follows:
Using subjective scales to assess visual comfort while using objective measures to reach more accurate conclusions was a recommended method [14].
Reviewer 4’ Major Comment No. 2:
- P4, L136: "All subjects were tested for both". Also please explain which tests were used for both (e.g. Landolt chart, or Ishihara plate test.)
Response:
Thank you for your useful advice. We have explained which tests were used in our paper.
The revised text is as follows:
Ishihara plate test was used to test both achromatic and chromatic visual acuity for all subjects [1], and the test results of 46 subjects all met the test standards.
[1] Ishihara S. The series of plates designed as a test for colour deficiency. Kanehara, 2001.
Reviewer 4’ Major Comment No. 3:
- Tables 2 and 5. Please include "Test statistic" as the title for the first column. Also "P" should be "p-value" This can also be applied to Tables 3 and 6.
Response:
Thanks for your suggestions. We have corrected the problems in the tables.
The revised text is as follows:
Table 2. Test statistic result.
|
CCT |
IL |
CCT*IL |
|
F |
7.09 |
9.22 |
0.504 |
|
p-value |
0.006** |
0.027* |
0.645 |
|
Partial η2 |
0.490 |
0.515 |
0.278 |
**p<0.01, *p<0.05; IL stands for illumination level.
Table 5. Test statistic result.
|
CCT |
IL |
CCT*IL |
|
F |
7.09 |
3.22 |
2.67 |
|
p-value |
0.005** |
0.011* |
0.689 |
|
Partial η2 |
0.477 |
0.298 |
0.033 |
**p<0.01, *p<0.05; IL stands for illumination level.
Table 3. Bonferroni-corrected pairwise comparison result.
CCT (K) |
p-value |
IL (lx) |
p-value |
||
3300 |
4300 |
0.279 |
300 |
500 |
0.002** |
3300 |
5300 |
0.001** |
300 |
750 |
0.018* |
4300 |
5300 |
0.037* |
500 |
750 |
0.289 |
**p<0.01, *p<0.05; IL stands for illuminance level.
Table 6. Bonferroni-corrected pairwise comparison result.
CCT (K) |
p-value |
IL (lx) |
p-value |
||
3300 |
4300 |
0.058 |
300 |
500 |
0.026* |
3300 |
5300 |
0.025* |
300 |
750 |
0.166 |
4300 |
5300 |
0.179 |
500 |
750 |
0.021*s |
*p<0.05; IL stands for illumination level.
Reviewer 4’ Major Comment No. 4:
- Figure 6. Please provide a small space between the values and the upper confidence interval in the plot, so that they do not overlap with each other. Similarly, this can be applied to the text labels in Figure 7.
Response:
Thanks for your valuable suggestions. And we have provided a small space between the values and the upper confidence interval in Figure 6. and Figure 7.
The revised figures are as follows:
Figure 6. Different PSD means according to CCTs and illumination levels; The vertical line represents the standard error.
Figure 7. Different mean error number of NOGO tasks according to CCTs and illumination levels and mean comfort score; The vertical line represents the standard error.
Reviewer 4’ Major Comment No. 5:
- Please label to primary and secondary y-axes for Figure 7. The error bars could also be edited (e.g. by using different colors and/or thicknesses), so that readers can distinguish between which belongs to the bars and which belongs to the line for each lighting condition on the x-axis.
Response:
Thanks for your suggestions. We have labeled to primary and secondary y-axes for Figure 7, and we have re-edited the error bars and the figures in different colors.
The revised figure is as follows:
Figure 7. Different mean error number of NOGO tasks according to CCTs and illumination levels and mean comfort score; The vertical line represents the standard error.
